# Feasibility of Virtually Delivering Functional Fitness Assessments and a Fitness Training Program in Community-Dwelling Older Adults

**DOI:** 10.3390/ijerph20115996

**Published:** 2023-05-30

**Authors:** Christian Thompson, Kathryn N. Porter Starr, Elizabeth Chmelo Kemp, June Chan, Emily Jackson, Justin Phun

**Affiliations:** 1Department of Kinesiology, University of San Francisco, San Francisco, CA 94117, USA; 2Department of Medicine, Duke University School of Medicine, Durham, NC 27710, USA; 3Geriatric Research Education and Clinical Center, Durham VA Health Care System, Durham, NC 27710, USA; 4Vivo, Durham, NC 27710, USA

**Keywords:** functional fitness, older adults, exercise, assessment, balance, muscle strength, cardiovascular fitness, telemedicine

## Abstract

The COVID-19 pandemic limited older adults’ access to preventative and diagnostic services and negatively affected accessibility to age-appropriate exercise programs. The purpose of this study was to assess the feasibility of conducting guided virtual functional fitness assessments before and after participation in an 8-week virtual, live fitness program (Vivo) designed for older adults. It was hypothesized there would be no significant difference between in-person and virtual functional fitness assessments and function would improve following the program. Thirteen community-dwelling older adults were recruited, screened, and randomly assigned to in-person-first or virtual-first fitness assessment groups. Validated assessments were delivered using standardized scripts by trained researchers and included Short Physical Performance Battery (SPPB) balance, a 30 s Chair Stand Test, 8 Foot Up-and-Go Test, 30 s Arm Curl Test, and 2 min Step Test. The eight-week, twice-a-week live virtual fitness program involved cardiovascular, balance, agility, Dual-Task, and strength training. Results showed no significant differences between all but one assessment measures, and several measures improved following the eight-week program. Fidelity checks demonstrated the high fidelity of program delivery. These findings illustrate that virtual assessments can be a feasible method to measure functional fitness in community-dwelling older adults.

## 1. Introduction

In December of 2019, the SARS-Cov-2 virus, better known as COVID-19, rapidly spread across the world. The relationship of morbidity and mortality and COVID-19 among older adults has been thoroughly researched. Similar to the United States, the countries of Korea and Italy documented that over 80% of the deaths due to COVID-19 were in people aged ≥ 60 years [1]. The aging process affects many physiological processes, particularly the immune system, which leads to higher inflammatory responses to antigens and decreased effectiveness to suppress infections [2]. Moreover, a recent study found older adults to be susceptible to more serious forms of COVID-19 due to pre-existing chronic conditions, such as hypertension, pulmonary disease, and diabetes mellitus [3]. In the United States, 80% of adults aged ≥60 years are classified as having one chronic condition, and 50% have two chronic conditions [4]. Thus, with older adults having a higher likelihood of having a chronic disease, morbidity and mortality due to COVID-19 increases.

In addition to the COVID-19 pandemic, aging leads to many functional declines, including losses of muscle mass and muscle performance, characterized as a loss of muscle strength and power [5]. These physiological declines often lead to functional impairments that threaten the ability of older adults to live independently [5]. Fortunately, resistance training has been consistently demonstrated as an effective means of increasing muscle mass and maintaining muscle performance with aging [6,7,8]. Previous research supports the benefit of physical activity in slowing the physiologic changes associated with aging by promoting psychological and cognitive well being, managing chronic diseases, reducing the risk of physical disability, and increasing longevity [9,10].

Despite the well-known benefits of resistance training, only an estimated 8.7% of older adults in the United States meet the recommended amount of resistance training on a weekly basis [11]. Older adults commonly report several barriers to participating in resistance training activities, such as safety; fear; health concerns, including pain or fatigue; and a lack of social support [12]. These barriers became more pronounced by the social distancing and social isolation recommendations resulting from COVID-19, which, unfortunately, has resulted in many negative health outcomes in older adults [13,14]. Additionally, a recent COVID-19 study reported a 72% reduction in the likelihood of commuting to participate in physical activity and a 140% increase in physical inactivity [15]. Furthermore, the lack of access to in-person physical activity programs has also impacted the ability to assess physical function status, an important tool used to measure physiological reserves and identify individuals at risk for falls, disability, and frailty [16].

Fortunately, virtual fitness programs have provided opportunities to participate in physical activity from a person’s home and provide older adults with an accessible option to exercise that they may otherwise not have pursued. Costs for virtual fitness programs (i.e., internet access plans, webcams for computers/tablets) are similar to that of a fitness club (i.e., membership costs). However, there are limited published studies on the use of virtually delivered functional fitness assessments; yet, they have been promising [17,18,19]. Ogawa and colleagues found that virtual functional fitness assessments have high reliability and support the utility of these tests to identify clinically relevant improvements in managing conditions such as diabetes and depression [17]. Additionally, although virtual assessments require older adults to modify their home environments to be assessed effectively, there is evidence that they can be conducted safely [19].

Therefore, the purpose of this study was to assess the feasibility of conducting virtual functional fitness assessments and an eight-week virtual fitness training program specifically developed for community-dwelling older adults with low-to-moderate activity levels (www.teamvivo.com, accessed on 15 May 2023). It was hypothesized that there would be no significant differences between virtual and in-person functional fitness assessments at matching timepoints, and functional improvements would be seen following participation in the eight-week virtual fitness training program.

## 2. Materials and Methods

### 2.1. Participant Recruitment and Enrollment

Approval for human subjects research was obtained by the Institutional Review Board of the lead researcher’s university prior to the study. A recruitment flyer was distributed to members of a membership-based village organization that assists older adults in maintaining independent living. Inclusion criteria included being 60 years of age and older, community-dwelling, not participating in resistance training twice a week during the past 6 months, and having a laptop or desktop computer with a camera and WiFi. Due to this being a sample of convenience, sample size for statistical power was not calculated for this study, but rather determined by interest and availability of members of the organization during the narrow time window of implementation. Additional participant recruitment and additional program offerings were not possible so it was deemed that sample size would be based on availability rather than statistical power.

Prospective participants who indicated interest were scheduled for a two-step screening process, which started with a virtually hosted consent visit with the research team, where informed consent was obtained. The Physical Activity Readiness Questionnaire (PAR-Q+) was used to determine if it was safe for an individual to participate in the exercise program [20]. If any PAR-Q+ question was answered with a YES, then a medical release was requested. If a clinician did not support the inclusion of the participant, then the participant would have been excluded from participation. Two participants were asked to obtain written medical clearance to participate in the study due to medical history collected on the first page of the PAR-Q+.

Following the screening meeting, the participant was officially enrolled into the study and was randomized into one of two groups, which determined the order each participant would receive their pre-test and post-test assessments: virtual first, where participants completed their initial functional fitness assessment battery through a virtual appointment with a trained research staff member or in-person first, where participants completed their initial functional fitness assessment battery during an in-person appointment with a different set of trained research staff. Participants completed the second of their functional fitness assessment batteries between 48 and 72 h after the other assessment’s delivery format. The same order of assessment format for each participant was performed at post-test.

Despite modifications made in terms of the set up and instructions to accommodate setting, ensuring participant understanding and safety, the virtual and in-person assessments that the participant performed were the same. All of the assessments were carried out using the same standardized script and were executed in the same order. The in-person and virtual assessments are explained in greater detail below. Participation in all phases of the project was voluntary and participants could drop out of the study at any time and for any reason.

In total, 16 participants were initially enrolled into the study. Two did not finish the intervention period and one only completed 50% of the exercise sessions. Thirteen participants completed at least 70% of the exercise sessions and were assessed at both pre-test and post-test timepoints. Participants included ten females and three males with a mean age of 82.56 (±6.96) years and mean weight and height of 67.49 (±20.93) kg and 1.68 (±0.1073) m, respectively. Body mass index mean was 23.51 (±4.81) kg/m^2^.

### 2.2. In-Person Assessment Process

Participants were scheduled for a 60 min assessment appointment in a multi-purpose room in a community center where the research team delivered all assessments following standardized communication scripts. Prior to the assessment appointment, the room was arranged for testing with consistent room temperature and lighting. Testing stations in the room were laid out in a sequenced manner around the perimeter of the room. Participants started by completing height and weight measurements using a calibrated scale and a wall stadiometer. Participants then progressed through the functional fitness assessments in the following order: SPPB balance tasks (1 trial for feet together, staggered stance (1 right-foot-forward and 1 left-foot-forward trial) and tandem stance conditions (1 right-foot-forward and 1 left-foot-forward trial), 30-Second Chair Stand Test (1 trial), 8 Foot Up-and-Go Test (Single Task—2 trials), 8 Foot Up-and-Go Test (Dual Task—1 trial), 30 s Arm Curl Test (1 trial), and 2 min Step Test (1 trial). See Table 1 for a detailed list of the assessments performed. Participants were allowed up to three minutes of rest while transitioning from one assessment station to another, and researchers maintained continuous communication with participants to identify any adverse responses to the exercise testing. 

### 2.3. Virtual Assessment Process

Participants were virtually assessed via Zoom by different members of the research team than those who completed the in-person assessments in the same order that assessments were completed in the in-person format. Height and weight were not measured virtually. Researchers utilized the same commonly used, validated assessments listed above to an online administration at the start of COVID-19 with minimal disruption. Each assessment session lasted approximately 60 min in duration. Prior to the virtual assessment, participants were emailed standardized instructions, including equipment needed for the assessments and setup instructions detailing directions for each assessment. Participants were mailed a testing kit, which included a tape measure to accurately measure the distance for the 8 Foot Up-and-Go Test and to measure the midpoint of thigh for the 2 min Step Test, and two resistance bands to perform the 30-Second Arm Curl Test in case dumbbells were not available. At the beginning of the assessment, the assessor ensured proper positioning of the computer and camera, including lighting and sound, and then surveyed the space for trip hazards, ensuring a safe environment for the assessment and for the fitness program. The computer video was positioned so that the assessor could observe the participant performing the full range of motion for each movement, count the number of repetitions and/or keep time, and ensure participant safety. Demonstrations of the movements were performed by the assessor when the assessments were not performed correctly. Participants began the virtual assessment by being guided by a trained assessor on the research team through the SPPB balance tasks (1 trial for feet together), staggered stance (1 right-foot-forward and 1 left-foot-forward trial) and tandem stance conditions (1 right-foot-forward and 1 left-foot-forward trial), 30 Second Chair Stand Test (1 trial), 8 Foot Up-and-Go Test (Single Task—2 trials), 8 Foot Up-and-Go Test (Dual Task—1 trial), and 30 s Arm Curl Test (1 trial), where females were encouraged to use a 5 lb dumbbell and males an 8 lb dumbbell. If dumbbells were not present, a resistance band of light or medium resistance was used. Lastly, participants completed the 2 min Step Test (1 trial). Vital signs were not performed during this test. This test required participants to find a space in the house in front of a wall or door where they had enough room to march in place. Using detailed instructions, including pictures, and with oversight from the assessor on camera to confirm proper measurement while on camera, the participant measured the midpoint of their thigh, half way between the top of their iliac crest and the middle of the patella. A piece of tape was placed on the participant’s thigh, indicating the midpoint. Then, the participant walked over the flat surface (wall or door), placing the thigh with the piece of tape nearest to the surface. The participant then transferred the piece of tape from their thigh to the wall at the same height indicating the midpoint of the participant’s thigh and the height at which to lift the knee during the assessment. The assessor used a handheld counter to count the number of times the right knee was lifted to the height of the tape mark during the two minutes. Additional assessment details can be found in Table 1.

### 2.4. Implementation of the Virtual Fitness Training Program

The virtual fitness training program was delivered by Vivo (www.teamvivo.com, accessed on 15 May 2023), a digital fitness solution company that provides live and interactive small group fitness training classes designed specifically for older adults. Different from its competitors, Vivo’s live format allows participants and trainers to interact in real time during the workout via video conference (i.e., Zoom meetings). This allows the trainers to demonstrate exercises, observe form, and provide safety oversight, exercise modifications, and support to the participants, fostering a safe and engaging exercise environment. Considering these benefits and that the population was older and less experienced with exercise, members of the research team chose Vivo as the intervention. It should also be noted that members of the research team are advisors to Vivo and have contributed to the design of Vivo’s exercise training programs.

The 8-week fitness program consisted of two 45 min sessions per week. Sessions were delivered via Zoom. Each session consisted of a 15 min warm up involving cognitive dual tasks, dynamic mobility exercises for the ankle, hip, spine, and shoulders, and balance exercises; three sets of three to five strength- or cardio-based exercises; and a 10 min cool down involving full-body dynamic stretching. Each exercise included variations ranging from level 1 to level 4 for participants to choose based on their comfortability or fitness level, and this allowed trainers to provide an individualized experience to each participant. The American College of Sports Medicine recommends that activity intensity for older adults be defined relative to an individual’s fitness within the context of perceived physical exertion [9]. As such, the fitness program intensity used the Vivo Intensity Protocol (VIP). The VIP is a subjective tool designed to help participants understand how hard they should expect to work during any given session. For example, a strength workout of three sets of eight repetitions should not feel exhausting and similarly, a high intensity interval training workout should not be performed at a leisurely pace. The VIP scale uses colors to help members understand their expected workout intensity, ranging from blue (light) to red (very hard). This coupled with the Talk Test is an easy way for members and trainers to assess workout intensity [25]. The 8-week fitness program was delivered by certified trainers who specialize in strength training for older adults, including functional and corrective movements. All study personnel (i.e., researchers, trainers, and research assistants) were trained and certified in CPR and First Aid and were proficient at monitoring the safety at all times. Safety during the virtual assessment and the exercise sessions was paramount especially since the trainer is not in the same physical space as the participant. The trainer observes the participant’s physical space for safety concerns and their physical appearance during each workout. Throughout the workout, participants reported their intensity level and symptomology (if any) so that the trainer could monitor and modify their workout as needed. Although CPR/First Aid are unable to be administered virtually, Vivo instructors are capable of activating an emergency protocol (e.g., advising a 911 emergency call) should an adverse incident arise. This study was voluntary and participants could drop out of the study at any time and for any reason. No adverse events were reported during the study.

### 2.5. Assessment of Exercise Fidelity

Fitness training program fidelity was monitored via four live or recorded exercise sessions. Researchers evaluated the class elements based on an observation checklist in order to assess the effectiveness of program delivery of each of the trainers. Fidelity checks included critiques of the trainers’ level of preparedness in gathering materials prior to the session; the clarity of their overview of the workout; effective delivery of the warmup, workout, as well as cool down; and their positive encouragement of participants during the session. Researchers determined a final fidelity score out of 26 based on the total number of boxes checked. Each of the four exercise classes were checked for fidelity by two independent observers and scored above a 22 on both checklists, indicating excellent fidelity.

### 2.6. Statistical Analysis

Descriptive statistics were calculated to summarize all dependent variables and two-tail dependent t-tests were conducted to determine if differences existed between in-person and virtual assessment variables. Additionally, one-tailed dependent t-tests were performed to determine if differences existed between pre-test and post-test timepoints for in-person functional fitness assessment measures. A probability value of ≤0.05 was used to determine significance. Based on calculated attendance, only participants who attended 70% or more of the training sessions were included in the data analyses.

## 3. Results

### In-Person and Virtual Functional Fitness Assessments

In-person and virtual functional fitness assessment measures for both pre- and post-test are shown in Table 2. At pre-test, the number of knee raises completed during the 2 min. Step Test differed significantly (*p* = 0.02) between the in-person and virtual fitness functional assessment (73.69 ± 21.91 versus 55.08 ± 22.95 steps, respectively), while no differences were detected between the other assessment measures. At the 8-week post-test, no significant differences were found between the in-person and virtual assessments delivery of any functional fitness assessment.

Significant differences were found in several of the in-person functional fitness assessments when comparing pre-test to post-test timepoints. Both lower and upper-body strength improved as measured by the 30 s Chair Stand Test (11.08 ± 2.25 versus 13.15 ± 2.12 repetitions, *p* = 0.00, effect size = 0.92) and the 30 s Arm Curl Test (14.69 ± 4.46 versus 17.38 ± 4.01 repetitions, *p* = 0.00, effect size = 0.67). The 30 s Arm Curl Test measures arm strength, where more repetitions indicate greater strength. The minimal detectable change for the Arm Curl Test has been reported at 2.3 repetitions with currently no reported minimal clinically important difference; with that being said, both the in-person and virtual Arm Curl Test showed improvements of two repetitions, indicating meaningful improvements in arm strength [26]. Lower-body strength was measured by the 30 s Chair Stand Test, where a greater number of chair stands completed in 30 s indicates greater leg strength. For this assessment, both the minimal detectable change and minimally clinically important difference has been reported as two repetitions [26,27]. Study results indicate only the 30 s Chair Stand Test reached the threshold of meaningful change, while the Arm Curl Test approached the threshold of meaningful change. Effect size for both the 30 s Chair Stand Test and the 30 s Arm Curl Test was above 0.5 and therefore considered large. Agility and dynamic balance improved both in a Single-Task 8 Foot Up-and-Go Fast Speed condition (7.53 ± 1.71 versus 7.28 ± 1.62 s, *p* = 0.04, effect size = 0.14) and Dual-Task condition (10.70 ± 2.38 versus 9.95 ± 1.97 s, *p* = 0.05, effect size = 0.38). In this study, neither the in-person or the virtual assessment showed the minimal detectable change of 1.4 s for the 8-Foot Up-and-Go Test [25]. Additionally, the 2 min Step Test demonstrated a significant improvement (73.69 ± 21.91 versus 83.92 ± 23.10 steps, *p* = 0.03, effect size = 0.46) for in-person and virtual assessments, indicating improved aerobic capacity following the 8-week virtual fitness training program. The effect sizes of the 8-Foot Up-and-Go Test and 2-Minute Step Test were below 0.5 and therefore were considered small. None of the static balance assessments improved significantly from pre-test to post-test.

## 4. Discussion

Functional fitness measures are commonly used in clinical and research settings to determine one’s functional abilities. With COVID-19 and the need to socially distance and isolate, alternative ways to conduct these measurements were necessary, as coming into a clinic or a research facility may not have been an option. While the research on the validity of virtual assessments compared to in-person assessments has been limited, our primary finding supports that virtual assessments can successfully obtain fitness measures for participants. In-person assessment may be the gold standard of fitness testing, but the advantages of virtual assessments should not be overlooked. Some of the benefits include reduced travel time to/from a clinic or research facility and decreased anxiety of visiting a hospital or facility that may be crowded or difficult to navigate. Live virtual assessments still are 1:1 interactions between the assessor and the participant and give uninterrupted time for the participant to ask questions, build rapport between the parties, and give the assessor an opportunity to observe the participant’s home environment, which may reveal potential hazards or suboptimal arrangements [28,29]. This would not be possible with in-person assessments. 

The disadvantages of virtual assessments include the difficulty of ensuring the exact procedures. Detailed instructions help to describe the assessments, provide illustrations and step-by-step set-up procedures, and provide guidance on how the assessment will be performed. The assessor must be well-trained to ensure proper set-up and closely monitor the participant’s movement throughout the assessment for proper scoring. However, with preparation, this study clearly demonstrates that virtual assessments are feasible and can be conducted with high success rates.

When comparing the in-person with the virtual assessment variables at matching timepoints, this study found that only the 2 min Step Test at pre-test was significantly different. While the setup and instructions for the assessments were the same, there were significantly more steps taken in the in-person modality to ensure proper set up and understanding than the virtual modality. There has been excellent intra- and inter-reliability demonstrated when previous research has delivered the 2 min Step Test in person [18,30,31]. Since the post-test timepoint did not reveal significant differences on this metric, additional research may be needed to further evaluate the virtual delivery of the 2 min Step Test. On the whole, this study supports the evidence that virtual assessments are a feasible and valid method of assessing functional fitness assessments when standardized procedures are used.

During COVID-19, virtual exercise training provided a method of keeping older adults physically active and engaged. As previously mentioned, the COVID-19 pandemic led to a decrease in physical activity in older adults due to several barriers, but especially due to the lack of in-person fitness training opportunities, as gyms and other facilities were closed. The results of this study show it is feasible to deliver a virtual fitness training to community-dwelling older adults and achieve improvements in overall functional fitness from baseline to post-assessment measurements. Virtual fitness training programs have benefits that should not be overlooked, including increased accessibility and convenience, social engagement through camera and microphone features on video meeting platforms such as Zoom, and individualized attention from a certified fitness professional. Participants indicated that the program of choice for this study, Vivo (www.teamvivo.com: accessed on 15 May 2023), was effective, with 12 of the 13 participants reporting being either “Satisfied” or “Very Satisfied” with the training program. Three of the participants joined Vivo following the study as subscribed members. While in-person training may have similar benefits, the aims of a training program and the goals of the participants should be considered when deciding between virtual training and in-person fitness training.

A recent meta-analysis examining training programs for functional fitness in older adults, with a wide range of durations ranging from six months to over two years, showed overall improvements in muscle strength and lower-body strength [21]. It should be noted that the duration of this study (8 weeks) was significantly shorter compared to other interventions [7,8]; despite this, similar improvements in overall muscle strength and in lower-body strength were seen. In regard to agility, a previous review attributed significant improvements to the multi-modal training styles with mobility-focused exercises in community-dwelling older adults [32]. In this study, the measure used for agility and dynamic balance was the 8 Foot Up-And-Go Test, along with the 30-Second Chair Stand Test, which both showed significant improvements after 8 weeks of Vivo. Cardiovascular fitness, as measured by the 2-min Step Test, showed significant improvements, which also has been confirmed by previous studies. In a similar study of progressive training during the COVID-19 pandemic, aerobic fitness, along with muscular fitness, balance, and agility, significantly improved after a 12-week exercise program, demonstrating that exercise programs can elicit adaptations to exercise [33]. However, similar improvements in balance from this previous study were not observed in the present study. One explanation for this may be that a duration of 11–12 weeks of balance-specific training is needed to elicit the most effective improvements in balance in healthy older adults [31]. 

Findings from this study suggest that virtual functional fitness measures are feasible in older adults and are comparable to in-person functional assessments. While safety of participants and technological issues are primary concerns with virtual programming, this study resulted in minimal technological problems and no reported adverse events. A limitation of this study was that the program was delivered to a sample of older adults living in a higher-income suburb of a large American city. There are known to be associations between access to technology, utilization of virtual healthcare, advancing age, rural residency, and lower socioeconomic status [34]. Future research should examine similar virtual interventions in diverse populations in order to generalize the findings.

Additionally, the small sample size and short duration of the study may explain why some functional measures approached clinically meaningful changes and others did not. Future studies of larger size and longer duration should explore factors that may influence meaningful changes in these assessments. Another limitation of this study may have been related to the participant selection. Those who volunteered for the virtual fitness training program, may have been more confident in their technology abilities and therefore, more likely to be successful in the study. Future studies may consider expanding the inclusion criteria to include those with a wide range of technology abilities to help determine the successful delivery of virtual assessments and virtual fitness training in older adults.

## 5. Conclusions

The COVID-19 pandemic provided an opportunity to determine the feasibility of utilizing virtual functional fitness assessments when in-person functional fitness assessments were not an option. This study successfully delivered virtual functional fitness assessments, showing no difference between them and the in-person functional fitness assessments except for the baseline 2 min Step Test measurement. This study also showed functional improvements in several aspects of functional fitness following 8 weeks of the virtual fitness training program.

## Figures and Tables

**Table 1 ijerph-20-05996-t001:** Functional fitness assessment measures.

Functional Fitness Assessment Protocols (Baseline and Post-Intervention)	
Balance component of the SPPB [21]	Static steady-state balance assessment of three stances (side-by-side, semi-tandem, tandem) was performed on a firm floor surface and required participants to hold their position for up to 10 s each. One-leg stands for left and right foot were held for up to 60 s. Side-by-side stand had the participant standing with their feet side by side. Semi-tandem stand had the participant touching the side of the heel of one foot to the big toe of the other foot. Tandem stand had the heel of one participant’s foot in front of and touching the toes of the other foot. Either foot was allowed to be put forward. One-leg stand had the participant standing on one foot and raising their other foot off the ground two inches.
30 s Chair Stand Test to quantify lower-body strength [22]	The number of stands completed in 30 s. The participant sat on the edge of a standard 17” seat height chair with arms folded across the chest and was instructed to perform repeated sit to stand repetitions at the most rapid rate possible. Feedback was given to ensure that participants fully stood up and made contact with the seat of the chair at the bottom of the repetition.
8 Foot Up-and-Go Test for agility and dynamic balance [23]	The number of seconds to get up from a seated position, walk 8 feet forward (2.44 m), turn around a small cone, and return to the chair and take a seated position. The participant began seated in the middle of the chair with their feet flat on the floor and hands on their thighs. One foot was slightly in front of the other, and the participant’s torso was slightly leaning forward. Test was performed four times. The first two trials, the participants walked as fast as they safely could. The third trial was executed at the participant’s normal walking pace. The last trial had the participant naming all of the words they could think of that began with a specific letter (e.g., F, A, S) at their normal walking pace. Dual-Task cost was calculated using the results from trials three and four: ([(Dual-Task performance − Single-Task performance)/Single-Task performance)] × 100) [24].
30 s Arm Curl Test to quantify upper-body strength [22]	The number of arm curls completed in 30 s on the right and left side using a dumbbell with a suitcase grip. The participant began seated in a 17” standard chair with an arm positioned vertically down beside the chair. Gradually the participant turned the palm up as the arm curled through the full range of motion and was lowered through the full range of motion returning to the starting position. To be counted, the participant’s arm had to be fully bent and then fully extended at the elbow. Males were tested with an 8 lb dumbbell and females were tested with a 5 lb dumbbell.
2 min Step Test for aerobic endurance [22]	The number of right knee raises completed in two minutes while marching in place and raising each knee to a point midway between the patella (kneecap) and iliac crest (top hip bone). Participants were instructed to march at their maximum safe speed and were instructed to take a short break if needed during the 2 min period. No vital signs were collected during this test.

**Table 2 ijerph-20-05996-t002:** In-person and virtual assessment descriptive statistics for all subjects (n = 13).

	Pre-Test		Eight-Week Post-Test	
Assessment	In-Person Mean ± SD	Virtual Mean ± SD	In-Person Mean ± SD	Virtual Mean ± SD
Side-by-Side Stance (s)	10 ± 0	10 ± 0	10 ± 0	10 ± 0
Semi-Tandem Stance (s)	9.92 ± 0.27	10 ± 0	9.68 ± 1.216	10 ± 0
Tandem Stance (s)	8.82 ± 2.87	9.23 ± 2.77	8.84 ± 2.47	8.61 ± 3.53
Single Leg Stance Right (s)	22.75 ± 23.95	12.67 ± 10.59	22.69 ± 24.32	19.15 ± 18.98
Single Leg Stance Left (s)	17.80 ± 18.94	11.64 ± 11.36	23.75 ± 34.90	19.89 ± 18.99
30 s Chair Stands (reps)	11.08 ± 2.25	11.62 ± 2.40	13.15 ± 2.12 *	12.86 ± 2.48
8 Foot Up-and-Go Fast (s)	7.53 ± 1.71	6.96 ± 1.43	7.28 ± 1.62 *	6.55 ± 2.14
8 Foot Up-and-Go Usual (s)	9.54 ± 2.28	8.75 ± 2.16	9.04 ± 1.54	8.04 ± 1.44
8 Foot Up-and-Go Cog (s)	10.70 ± 2.38	9.47 ± 2.65	9.95 ± 1.97 *	9.87 ± 2.52
30 s Arm Curl Right (reps)	14.69 ± 4.46	16.15 ± 4.14	17.38 ± 4.01 *	19.29 ± 4.94
2 min Step Test (steps)	73.69 ± 21.91	55.08 ± 22.95 ^†^	83.92 ± 23.10 *	68.86 ± 24.52

Note: * indicates significant difference (≤0.05) between pre-test and post-test measurements. ^†^ indicates significant difference (≤0.05) between in-person and virtual assessments at matching timepoint.

## Data Availability

Data and data analyses are available and may be reviewed by following this Google Drive link and then requesting access by contacting Christian Thompson at cjthompson@usfca.edu: https://drive.google.com/drive/folders/1a5NgFX1J0izZ17IjbLBpRLM3aB9aPD84?usp=share_link, accessed on 10 May 2023.

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
