# Peer review of "Feasibility of Virtually Delivering Functional Fitness Assessments and a Fitness Training Program in Community-Dwelling Older Adults"

_ijerph, 2023, doi:10.3390/ijerph20115996_

Round 1

Reviewer 1 Report

The authors are addressing an important need with a potentially practical home-based fitness assessment and intervention program for elders. However, several important details are lacking in the manuscript.

Introduction

·      In-depth background information on the differential outcomes of COVID experienced by older versus younger people doesn’t seem necessary as a rationale for this investigation. That said, the assessment and development of functional strength, balance, etc. in older people is important for a number of health-related reasons independent of COVID. Also important is the need to make a virtual and safe version of assessment and intervention available during times when travel and use of a gym facility are limited.

·      What questions or challenges have been raised in the literature regarding the virtual administration of assessments or functional strength interventions in this population? There is only one citation (16) on the lack of access to virtual programs.

·      “Feasibility” needs to be defined for this study. Beyond a successful small trial in which it was possible to complete assessments and interventions as specified (which was largely accomplished in this study), it would be good to know the full range of costs associated with virtual versus in-person administration. Projected cost to consumer would matter too. Also, differences in participant satisfaction, intention to continue exercise, etc., would have been informative. 

Methods

·      More detailed information is needed on the inclusion and exclusion criteria. One or more participants were excluded during enrollment for having an “unstable health condition” (p. 2, line 93). What, if any, such conditions were formally listed as exclusion criteria?

·      Presumably important conditions for performing the in-person assessment included room temperature, lighting, exercise stations laid out in a sequence, calibrated scale and wall stadiometer. Those conditions were not addressed in the virtual assessment. Are they important? Unless provided, home scales are not calibrated.

·      Were vital signs assessed during the in-person testing?

·      I’m not totally clear on what Vivo is, beyond an exercise and coaching program for older people, with the VIP and Talk Test (which sound very helpful). I see that Vivo's delivery is via Zoom, so it isn’t a delivery system, as I understand it. Is there more to the package?

·      Instructors’ CPR and first aid training would appear to be of limited use in a virtual program, other than to (perhaps) detect a medical emergency and call for assistance.

·      Fidelity checks were conducted by one person scoring a checklist, as I understand it. I don’t see any independent verification (i.e., a comparison of two independent observers and an associated index of agreement.)

Results

·      Thank you for presenting clear pre- and post-results of the fitness measures.

·      Height and weight were taken, but weight/BMI status was not reported.

Discussion

·      How Vivo compares with and/or improves upon other available video-based programs for elders is not elaborated.

·      Again, feasibility should include data on what it would cost the provider and the consumer to use, and on safety measures.

·      What steps would be included in Vivo delivery in the case of a medical event?

Author Response

Thank you for the consideration of this manuscript and your input.  Below please find responses to each of your comments:

Introduction

  •     In-depth background information on the differential outcomes of COVID experienced by older versus younger people doesn’t seem necessary as a rationale for this investigation. That said, the assessment and development of functional strength, balance, etc. in older people is important for a number of health-related reasons independent of COVID. Also important is the need to make a virtual and safe version of assessment and intervention available during times when travel and use of a gym facility are limited.
    • We agree with that comment - there is a place for virtual training in the future of older adult fitness and this manuscript is intended to provide helpful information to readers that it is feasible to deliver virtual assessments as long as detailed protocols are in place.
  •     What questions or challenges have been raised in the literature regarding the virtual administration of assessments or functional strength interventions in this population? There is only one citation (16) on the lack of access to virtual programs. 
    • We have added 2 additional references detailing the effective use of virtual fitness assessments for older adults with adequate space and set-up modifications made.  Additionally, virtual assessments can be clinically meaningful for tracking participant's progress.  1.  E Yilmaz, B Akinci, G Utku, E Erdinc, I Atmaca, EN Gurlek, An online functional assessment experience in individuals over 65+ during Covid 19 pandemics: physiotherapist opinion & participant opinion, European Journal of Cardiovascular Nursing, Volume 20, Issue Supplement_1, July 2021, zvab060.140, https://doi.org/10.1093/eurjcn/zvab060.140 2 and Ogawa EF, Harris R, Dufour AB, Morey MC, Bean J. Reliability of Virtual Physical Performance Assessments in Veterans During the COVID-19 Pandemic. Arch Rehabil Res Clin Transl. 2021 Jul 21;3(3):100146. doi: 10.1016/j.arrct.2021.100146. PMID: 34589696; PMCID: PMC8463460.
    • esponse here: response in comment. Other barriers to lack of access to virtual programs include; limited knowledge of the programs- most programs are referred by healthcare providers. Vivo is still new and breaking into the referral space takes time. Lack of space at home- we need no more space that it takes to hold arms out straight to sides and turn around (about 7 feet). Lack of equipment- Vivo sends welcome kits and exercise can be done with minimal equipment. 
  •     “Feasibility” needs to be defined for this study. Beyond a successful small trial in which it was possible to complete assessments and interventions as specified (which was largely accomplished in this study), it would be good to know the full range of costs associated with virtual versus in-person administration. Projected cost to consumer would matter too. Also, differences in participant satisfaction, intention to continue exercise, etc., would have been informative. 
    • Good suggestion.  Virtual exercise programs such as Vivo require stable, reliable internet ($50 USD/month) and a computer/tablet.  Most computers have cameras but a webcams can be purchased for $150 USD. Compared to in person gym members or personal training which costs gas/mileage to gym and monthly membership fees. 
      There are risks by doing assessments at home unassisted- mainly risk of injury, however performing live virtual assessments with trained personnel can help to reduce risk of injury.  Following the completion of this study, 12 of the 13 participants reported being satisfied or very satisfied with the program and 3 of the 13 participants joined Vivo as ongoing subscribing members.

Methods

  •     More detailed information is needed on the inclusion and exclusion criteria. One or more participants were excluded during enrollment for having an “unstable health condition” (p. 2, line 93). What, if any, such conditions were formally listed as exclusion criteria?
    • If any PAR-Q+ screening question was answered YES then a medical release was sought.  If the clinician had not support the inclusion of the participant he/she would have been excluded.
  •     Presumably important conditions for performing the in-person assessment included room temperature, lighting, exercise stations laid out in a sequence, calibrated scale and wall stadiometer. Those conditions were not addressed in the virtual assessment. Are they important? Unless provided, home scales are not calibrated.
    • The virtual assessments were delivered in the same order as the in-person assessments.  Body weights were not collected virtually.  

Response here:  

  •     Were vital signs assessed during the in-person testing?
    • No, however participants were in constant communication with the assessment team to ensure they were not having any symptoms of medical complications, such as chest pain or shortness of breath
  •     I’m not totally clear on what Vivo is, beyond an exercise and coaching program for older people, with the VIP and Talk Test (which sound very helpful). I see that Vivo's delivery is via Zoom, so it isn’t a delivery system, as I understand it. Is there more to the package?
    • Vivo is an online, live and interactive fitness program for older adults (Teamvivo.com). 
  •     Instructors’ CPR and first aid training would appear to be of limited use in a virtual program, other than to (perhaps) detect a medical emergency and call for assistance.
    • Here's more information:  Instructor qualifications are listed to further illustrate that exercise leaders were trained and certified fitness professionals.  Although no CPR/First Aid could be administered virtually, instructors with these qualifications are capable of detecting medical emergencies if they were to arise and constant communication with participants ensures any adverse event is rapidly identified and medical treatment is immediately sought
  •     Fidelity checks were conducted by one person scoring a checklist, as I understand it. I don’t see any independent verification (i.e., a comparison of two independent observers and an associated index of agreement.)
    • Clarification:  Fidelity checks were completed by 2 independent observers and compared after the reviews were completed.  All sessions indicated high fidelity.

Results

  •     Thank you for presenting clear pre- and post-results of the fitness measures.
  • Height and weight were taken, but weight/BMI status was not reported.
    • Response Here: BMI Mean: 23.51, SD 4.81, min 16.12, Max 34.11

Discussion

  •     How Vivo compares with and/or improves upon other available video-based programs for elders is not elaborated.
    • Vivo is an online, live and interactive fitness program for older adults (Teamvivo.com).  Each participant is actively engaged by a single instructor throughout each live class period.  There are no archived videos, it is all live fitness education.
  •     Again, feasibility should include data on what it would cost the provider and the consumer to use, and on safety measures.
    • Addressed above in cost breakdown
  •     What steps would be included in Vivo delivery in the case of a medical event? 
    • Further elaboration beyond the Methods Section:  Participants were encouraged to self-report adverse events to trainers and staff at any time. Additionally,  study staff closely monitored participants for any potential safety events occurring including but not limited to any unusual signs, symptoms, or changes in health status from when the study first started. The study staff, including the trainers, the assessment team and the customer service team are all carefully trained and have the expertise to safely conduct virtual strength training and physical function measures. This includes trainers who are certified in personal training and specialize in corrective exercise and an aging population.
      During any portion of the workout or assessment, if an individual can no longer be seen by the trainer/trained research staff and is unresponsive for more than one minute, falls, reports shortness of breath or an injury, then the trainer will stop the class, move additional participants to a breakout room and will initiate the emergency response protocol.

Reviewer 2 Report

The main purpose of the current study was to compare two different assessment procedures (in-person functional assessment and self-functional assessment) used to investigate the effects of a virtual functional fitness training program (programmed by the Vivo company) in community-dwelling older adults. The current study provides information beneficial to telerehabilitation that can be adapted for the new normal era. However, several points of the current study are unclear to the readers and need improvement.

Major points

Title:

It is beneficial to the readers to follow the article if the title can be rewritten according to the study’s main purpose or the main point the authors want to present.

Abstract: 

1.      The manuscript’s abstract has too many words (>200), which is not in line with the recommendation from the journal. Therefore, this part should be summarized, especially in the Background and Result sections.

2.      It would be beneficial to add one more keyword, e.g., “telerehabilitation.”

Introduction:

1.      Please provide information about the Vivo and explain why the 8-week program that the current study was selected.

Materials and Methods:

1.      In the subsection “2.1. Participant Recruitment & Enrollment”, please provide information about the ethical approvement of the study protocol. In addition, please provide information about the sample size calculation. Otherwise, if the current study did not calculate the sample size, please explain why.

2.      The subsection “3.1. Study Population and Retention” should be in the subsection ”2.1 Participants” of the Materials and Methods, and the details of participants’ fitness levels (mentioned in lines 172-173) should also be summarized in this part.

3.      I am unsure that the study design is a cross-over study. Please provide more information about the study design of the current study. In addition, providing a flowchart describing the experimental protocol will help the readers to understand the study more.

4.      In the subsection “2.2. In-Person Assessment Process”, please list the name of the assessments in the text rather than in the table and move Table 1 to locate at the end of this subsection.

5.      In the subsections “2.2 and 2.3.”, please provide the number of tests conducted for each functional assessment. In addition, are the same test sequences conducted for in-person assessment and self-assessment?

6.      In the subsection “2.4. Implementation of the Virtual Fitness Training Program”, please add the details of Vivo, e.g., company name, URL, or reference. According to Vivo, if the program is created for a small group of older adults, this may be the reason for the small sample size of the current study. Please clarify this point in subsection “2.1 Participants.”

7.      In the subsection “2.6. Data Analysis”, please provide information on data normality testing, the statistics used in the study, and the effect size calculations.

Results:

1.      The subsection “3.1. Study Population and Retention” should be in the subsection ”2.1 Participants” of the Materials and Methods.

2.      In the subsection “3.2. In-Person and Virtual Functional Fitness Assessments”, please add the p-value and effect size in the significant results, e.g., in lines 236-237, 248-249, and 252.

Discussion:

1.      The first paragraph should summarize the objectives and main findings of the study.

2.      According to the clinical implication, please suggest a further application. For example, the points (e.g., validity and reliability) should be aware, especially the balance tests (as shown in Table 2). In addition, are the authors concerned that the small sample size may make no significant difference between the two types of assessments (in-person functional assessment and self-functional assessment)?

Minor points

1.      In Table 1, please do not use the symbol #.  It would be easy to read if using the full name.

2.      In the subsection “2.6. Data Analysis”, using the word “Statistical analysis” is suitable since the term “Data analysis” is commonly used for data processing.

3.      In Table 2, please add the number of participants in the table caption (e.g., (n = …)).

4.      In lines 277-279, the reference is missing.

Author Response

Thank you for your review of our manuscript.  Your comments, suggestions and questions were quite helpful. Please see below for our responses.

The main purpose of the current study was to compare two different assessment procedures (in-person functional assessment and self-functional assessment) used to investigate the effects of a virtual functional fitness training program (programmed by the Vivo company) in community-dwelling older adults. The current study provides information beneficial to telerehabilitation that can be adapted for the new normal era. However, several points of the current study are unclear to the readers and need improvement.

Major points

Title:

It is beneficial to the readers to follow the article if the title can be rewritten according to the study’s main purpose or the main point the authors want to present.

  • We feel the title conveys both the ability (feasibility) to gather valid and reliable functional fitness data via virtual administration and that a virtually-delivered exercise program can have beneficial effects on developing functional fitness.  We would appreciate if the suggestion of a title change is reconsidered or if an alternative were suggested.

Abstract: 

  1. The manuscript’s abstract has too many words (>200), which is not in line with the recommendation from the journal. Therefore, this part should be summarized, especially in the Background and Result sections.
    1. Modified, thank you.
  2. It would be beneficial to add one more keyword, e.g., “telerehabilitation.”
    1. Done, thank you.

Introduction:

  1. Please provide information about the Vivo and explain why the 8-week program that the current study was selected.
    1. Vivo is an online, live and interactive fitness program for older adults (Teamvivo.com).  The 8-week exercise program selected was specifically designed for community-dwelling older adults of low to moderate levels of baseline functional fitness which reflected this sample.

Materials and Methods:

  1. In the subsection “2.1. Participant Recruitment & Enrollment”, please provide information about the ethical approvement of the study protocol. In addition, please provide information about the sample size calculation. Otherwise, if the current study did not calculate the sample size, please explain why.
    1. Approval of this study was obtained through the lead researcher's university Institutional Review Board for the Protection of Human Subjects
    2. Sample size was was not calculated for this study.  This study was a sample of convenience and was limited by interest and availability with a narrow time window.  Additional participant recruitment was not possible so it was deemed that sample size would be based on availability rather than statistical power.
  2. The subsection “3.1. Study Population and Retention” should be in the subsection ”2.1 Participants” of the Materials and Methods, and the details of participants’ fitness levels (mentioned in lines 172-173) should also be summarized in this part.
    1. Corrected. 
  3. I am unsure that the study design is a cross-over study. Please provide more information about the study design of the current study. In addition, providing a flowchart describing the experimental protocol will help the readers to understand the study more.
    1. Corrected, thank you for the suggestion
  4. In the subsection “2.2. In-Person Assessment Process”, please list the name of the assessments in the text rather than in the table and move Table 1 to locate at the end of this subsection.
    1. Assessments are now listed in the text and the table fully describing the assessments is now at the end of the subsection.
  5. In the subsections “2.2 and 2.3.”, please provide the number of tests conducted for each functional assessment. In addition, are the same test sequences conducted for in-person assessment and self-assessment?
    1. Number of trials has been updated and sequences were identical for in-person and virtual assessment (please understand that the virtual assessment was not a self-assessment - it was completed by a trained researcher who instructed, conducted and observed the participant doing the assessment according to standardized protocol)
  6. In the subsection “2.4. Implementation of the Virtual Fitness Training Program”, please add the details of Vivo, e.g., company name, URL, or reference. According to Vivo, if the program is created for a small group of older adults, this may be the reason for the small sample size of the current study. Please clarify this point in subsection “2.1 Participants.”
    1. Done, thank you.
  7. In the subsection “2.6. Data Analysis”, please provide information on data normality testing, the statistics used in the study, and the effect size calculations.
    1. Done

Results:

  1. The subsection “3.1. Study Population and Retention” should be in the subsection ”2.1 Participants” of the Materials and Methods.
    1. Done.
  2. In the subsection “3.2. In-Person and Virtual Functional Fitness Assessments”, please add the p-value and effect size in the significant results, e.g., in lines 236-237, 248-249, and 252.
    1. Done.

Discussion:

  1. The first paragraph should summarize the objectives and main findings of the study.
    1. Good suggestion.  Done.
  2. According to the clinical implication, please suggest a further application. For example, the points (e.g., validity and reliability) should be aware, especially the balance tests (as shown in Table 2). In addition, are the authors concerned that the small sample size may make no significant difference between the two types of assessments (in-person functional assessment and self-functional assessment)?
      1. Means of the virtual and in-person assessments that indicated no significant difference were within the minimal detectable change range for each assessment.  Therefore, not only were they statistically non-significant, they were considered similar based on detectable change.

Minor points

  1. In Table 1, please do not use the symbol #.  It would be easy to read if using the full name.
    1. Done
  2. In the subsection “2.6. Data Analysis”, using the word “Statistical analysis” is suitable since the term “Data analysis” is commonly used for data processing.
    1. Done
  3. In Table 2, please add the number of participants in the table caption (e.g., (n = …)).
    1. Done.
  1. In lines 277-279, the reference is missing.
    1. Updated.

Round 2

Reviewer 1 Report

The authors were responsive to the reviews and have produced a much-improved revised manuscript.

Author Response

Thank you for your feedback - we are pleased to know that the revisions were to your satisfaction.

Reviewer 2 Report

The current article has been improved on some points, but some critical issues are not concerned. As one of the readers, this article intended to promote the course and self-assessments (conducted by individual older adults). Therefore, in the discussion, the authors should be sincere in providing the points that practitioners or therapists are interested in applying their methods (both exercises and assessments), especially in self-assessment regarding advantages and disadvantages and essential points needed to be aware. This is because the data shown in Table, especially in the Pre-test are interesting such as Single Leg Stance Right (s) and Left (s) between in-person and virtual tests.

Moreover, the manuscript’s abstract has too many words (>200), which is not in line with the recommendation from the journal. Therefore, this part should be summarized, especially in the Background and Result sections. In addition, in the subsection “3.2. In-Person and Virtual Functional Fitness Assessments”, please add the p-value and effect size in the significant results, e.g., in lines 276-277, 288-289, and 292. In Table 1, please do not use the symbol #. It would be easy to read if using the full name.

Author Response

Thank you again for your helpful suggestions and comments.    We have incorporated these into our manuscript.  

Specifically - abstract is now less than 200 words

Effect sizes added to the Results section

Table 1 modified and eliminated "#" use

Discussion was further elaborated with 2 paragraphs.

Yellow highlights are for your consideration in the manuscript.

Thank you.